# Intranasal Treatment with Cannabinoid 2 Receptor Agonist HU-308 Ameliorates Cold Sensitivity in Mice with Traumatic Trigeminal Neuropathic Pain

**DOI:** 10.3390/cells13231943

**Published:** 2024-11-22

**Authors:** Simeng Ma, Yoki Nakamura, Suzuna Uemoto, Kenta Yamamoto, Kazue Hisaoka-Nakashima, Norimitsu Morioka

**Affiliations:** Department of Pharmacology, Graduate School of Biomedical and Health Sciences, Hiroshima University, 1-2-3 Kasumi, Minami-ku, Hiroshima 734-8553, Japan; d225717@hiroshima-u.ac.jp (S.M.); b180856@hiroshima-u.ac.jp (S.U.); m234539@hiroshima-u.ac.jp (K.Y.); hisaokak@hiroshima-u.ac.jp (K.H.-N.)

**Keywords:** cannabinoid receptor 2, cold sensitivity, intranasal treatment, post-traumatic trigeminal neuropathy, microglia

## Abstract

Post-traumatic trigeminal neuropathy (PTTN) is a sensory abnormality caused by injury to the trigeminal nerve during orofacial surgery. However, existing analgesics are ineffective against PTTN. Abnormal microglial activation in the caudal part of the spinal trigeminal nucleus caudal part (Sp5C), where the central trigeminal nerve terminals reside, plays an important role in PTTN pathogenesis. Therefore, regulating microglial activity in Sp5C appears to be an important approach to controlling pain in PTTN. Cannabinoid receptor 2 (CB_2_) is expressed in immune cells including microglia, and its activation has anti-inflammatory effects. The current study demonstrates that the repeated intranasal administration of CB_2_ agonist HU-308 ameliorates the infraorbital nerve cut (IONC)-induced hyperresponsiveness to acetone (cutaneous cooling). The therapeutic efficacy of oral HU-308 was found to be less pronounced in alleviating cold hypersensitivity in IONC mice compared to intranasal administration, indicating the potential advantages of the intranasal route. Furthermore, repeated intranasal administration of HU-308 suppressed the activation of Sp5C microglia in IONC mice. Additionally, pretreatment with the CB_2_ antagonist, SR 144528, significantly blocked the anti-nociceptive effect of repeated intranasal administration of HU-308 on cold hypersensitization in IONC mice. These data suggest that the continuous stimulation of CB_2_ ameliorates PTTN-induced pain via the inhibition of microglial activation. Thus, CB_2_ agonists are potential candidates for novel therapeutic agents against PTTN.

## 1. Introduction

Various studies have shown that more than 30% of the world’s population suffers from chronic pain [1]. Chronic pain imposes a significant psychological and socioeconomic burden on patients and reduces quality of life. However, it is resistant to existing analgesics, and thus, it is difficult to treat [2,3]. Therefore, there is an urgent need to search for new therapeutic approaches. The trigeminal nerve is the fifth cranial nerve, which is divided into three branches, ophthalmic, maxillary, and mandibular nerves, and it is involved in the sensation of the face, oral cavity, and nasal cavity [4]. Post-traumatic trigeminal neuropathy (PTTN) is a chronic pain condition caused by trigeminal nerve injury, which is associated with chronic hypoesthesia, allodynia, or both in the oral–facial region. PTTN develops in 3–7% of patients following dental or oral–facial surgery owing to trigeminal nerve injury [5]. However, non-steroidal anti-inflammatory drugs and opioid analgesics are not effective against the pain associated with PTTN [6,7]. Thus, other treatment modalities need to be developed.

Cannabinoid receptors are characterized as Gi protein-coupled receptors that are specifically activated upon the binding of tetrahydrocannabinol (THC), the primary psychoactive component of cannabis [8,9]. There are two subtypes of cannabinoid receptors, cannabinoid receptor 1 (CB_1_) and cannabinoid receptor 2 (CB_2_). CB_1_ is mainly expressed on neurons and has been reported to be involved in the analgesic and psychoactive effects of Δ9-THC, while CB_2_ is expressed on immune cells such as macrophages and microglia, and its activation is associated with anti-inflammatory, but not psychoactive, effects [9,10,11]. Thus, given that the stimulation of CB_2_ does not result in drug abuse, it is considered a promising target for the development of analgesic drugs. The interaction between the immune and nervous systems is important in neuropathic pain [12]. Microglia, the primary immune cells of the central nervous system, participate in the pathogenesis of chronic pain by releasing cytokines, chemokines, and other inflammatory mediators [13,14,15,16,17]. In our previous study, we found that abnormal microglial activation in the spinal trigeminal nucleus caudal part (Sp5C), a part of the medulla oblongata where the central trigeminal nerve terminals are located, plays an important role in the pathogenesis and exacerbation of PTTN [17,18]. However, whether CB_2_ is involved in PTTN-induced pain has not been investigated, to our knowledge.

To assess Sp5C microglial function in the brain using drugs, it is necessary to efficiently deliver them to that site. In previous studies, intranasal administration has been shown to efficiently deliver drugs to the olfactory bulb and the medulla oblongata via the olfactory and trigeminal nerves, respectively, offering advantages such as avoiding the blood–brain barrier, reducing dosage, and minimizing systemic side effects [19,20,21]. Therefore, intranasal administration is expected to deliver drugs to Sp5C microglia, which are abnormally activated in PTTN, and thus, this administration method may be a new therapeutic strategy against PTTN.

The present study focused on CB_2_ as a new therapeutic target for PTTN. We evaluated the effects of the intranasal administration of a CB_2_ agonist on pain-like behaviors and the hyperactivated state of Sp5C microglia in an infraorbital nerve cut (IONC) mouse model.

## 2. Materials and Methods

### 2.1. Animals

Male ddY mice (RRID: MGI:5558113; 8 weeks old) were obtained from Japan SLC, Inc. (Hamamatsu, Shizuoka, Japan), and housed under standard laboratory conditions (temperature: 23.5 ± 2 °C; humidity: 50 ± 10%; 12 h light/dark cycle). The animals were provided with standard rodent chow and water ad libitum. All experiments were conducted in accordance with the Guidelines for the Care and Use of Laboratory Animals established by the Japanese Pharmacological Society and approved by the Committee of Research Facilities for Laboratory Animal Science of Hiroshima University (approval numbers A19-65 and A20-163). All experimental procedures were performed through blinded experimenters. Experiments were conducted using groups of 5 mice to examine the effects of intranasal HU-308 administration (Figure 1A), oral HU-308 administration (Figure 1B), and pretreatment with SR 144528 prior to HU-308 administration (Figure 1C).

### 2.2. Infraorbital Nerve Cut Model as Post-Traumatic Trigeminal Neuropathy in Mice

In this study, the IONC model was used as the PTTN model [22]. Sodium pentobarbital (50 mg/kg, i.p., Cat. #: 26427-14; Nacalai Tesque, Kyoto, Japan) and 2% isoflurane (induction, 5%; maintenance, 2–3%; Cat. #: 099-06571; FUJIFILM Wako Pure Chemical Corporation, Osaka, Japan) were used to anesthetize ddY mice. A 3–4 mm incision was made in the skin overlying the left infraorbital nerve, which was then exposed and transected at two points 2 mm apart to induce neuropathic pain. For sham-operated controls, an identical incision was made but the nerve was left intact. Postoperatively, mice were closely monitored for recovery from anesthesia.

### 2.3. Acetone Test

The acetone test was used to measure cold hypersensitivity in mice [17,23]. Acetone sensitivity was evaluated at 7, 8, and 14 days post nerve injury (Figure 1). Prior to testing, the mice were habituated to a Plexiglas chamber (12 × 12 × 20 cm) for 15 min. Cold hypersensitivity was assessed 1, 3, 6, and 24 h after drug treatments. Fifteen microliters of acetone was applied to the left whisker pad, and the reaction time of the mouse was measured within 1 min. The response to the cold stimulation was defined as the duration that the mouse touched the left side of its face and rubbed its left cheek against the wall.

### 2.4. Drug Treatment

Mice were treated with intranasal administration (i.n.) or oral administration (p.o.) 8, 10, 12, and 14 days after the nerve injury. The temporal profile of analgesic effects was evaluated on day 8 after the initial drug administration for the single-administration group. For the repeated administration group, analgesic efficacy was assessed 14 days after the 4th drug administration (Figure 1). A pipette was used to administer 10 µL of the indicated solution(s) via intranasal administration. Considering the potential for drug reflux, the administration volume was decreased to 10 µL, which is lower than the typical volume of 20–30 µL [24]. Following intranasal administration, the animals were carefully monitored to ensure that no drug reflux occurred prior to behavioral testing. A mouse gavage needle was used to administer 100 µL of the indicated solution(s) via oral administration. Regardless of the administration methods, 30 nmol of CB_2_ agonist HU-308 (Cat. #: 90086; Cayman Chemical, Ann Arbor, MI, USA) and 100 nmol of CB_2_ antagonist SR 144528 (Cat. #: 9000491; Cayman Chemical, Ann Arbor, MI, USA) were used in this study. SR 144528, which has been reported to abolish the effects of HU-308 [25], was intranasally administered 15 min before each HU-308 administration.

### 2.5. Immunohistochemistry

Following behavioral testing, mice were euthanized with sodium pentobarbital (50 mg/kg) and 2% isoflurane and perfused transcardially with saline (50 mL), followed by 4% (*w*/*v*) paraformaldehyde (8% [*w*/*v*] paraformaldehyde dissolved 1:1 in 0.2 M phosphate buffer [pH 7.4], 20 mL). Brains were removed, post-fixed in 4% paraformaldehyde for 24 h, and cryoprotected in 30% (*w*/*v*) sucrose solution (60% [*w*/*v*] sucrose in 1:1 in 0.2 M phosphate buffer [pH 7.4]) at 4 °C for 3 days. Coronal brain sections (30 μm) were cut on a cryostat (CM3050S II, Leica Microsystems, Wetzlar, Germany). After air-drying, the sections were washed three times with 10 mM glycine/phosphate-buffered saline (PBS) at 5 min intervals and incubated in blocking solution (3% bovine serum albumin [BSA], 10% goat serum, 0.1% Triton X-100, and 0.05% Tween20) for 2 h at room temperature. The primary antibody used was rabbit polyclonal anti-ionized calcium-binding adapter molecule 1 (Iba-1; a microglia marker) antibody (1:500, Catalog # 019-19741, FUJIFILM Wako Pure Chemical Corporation, Osaka, Japan), diluted in 3% BSA in PBS. After incubation at 4 °C for three days, the primary antibody was removed, and the sections were washed six times with 0.1% BSA-PBS for 5 min per wash. Goat anti-rabbit IgG secondary antibody with Alexa FluorTM 555 (1: 500, Catalog # A21429, Thermo Fisher Scientific, Waltham, MA, USA) in 3% BSA-PBS was applied onto tissue sections on glass slides and incubated at 4 °C for 2 h. The antibody solution was then discarded, and the sections were washed six times with 0.1% BSA for 5 min per wash. Finally, the slides were sealed with a coverslip and observed using a BZ-900 Biorevo all-in-one fluorescence microscope (Keyence Corporation, Osaka, Japan). Individual Iba1-positive cells in the images were identified using CellProfiler^TM^ [26,27]. Briefly, for each cell, the cell number, the integrated fluorescence intensity of Iba1, and the average cell area were calculated using the MeasureObjectIntensity and MeasureObjectSizeShape module of Cell Profiler^TM^.

### 2.6. Statistical Analysis

Statistical analysis was performed using GraphPad Prism 7 (GraphPad Software, Boston, MA, USA). The results are presented as mean ± standard error of the mean (SEM). An unpaired t-test was used to compare two groups. The time course of the drug’s effects was analyzed using a two-way repeated measures (RM) analysis of variance (ANOVA) to confirm significance, followed by multiple comparisons using Tukey’s or Sidak’s method. Two-way ANOVA was used to compare the effects of different administration methods. Other data were analyzed by one-way ANOVA to confirm significance, followed by multiple comparisons using Tukey’s or Dunnett’s method. *p* < 0.05 was considered a significant difference. All the statistical details can be found in the Appendix A.

## 3. Results

### 3.1. Intranasal Administration of CB_2_ Agonist Alleviates Cold Hypersensitization in IONC Mice

Cold hypersensitivity is a common clinical manifestation in patients with trigeminal nerve injury [28]. Consistent with clinical findings, our experimental data demonstrate that IONC mice exhibited a significant increase in responsiveness to acetone, a cold stimulus, at 7–14 days post nerve injury (Figure 2A). Moreover, to assess the efficacy of CB_2_ agonist HU-308 in ameliorating this condition, we administered HU-308 intranasally to IONC mice in either a single or repeated dose regimen. While a single intranasal administration of HU-308 (30 nmol) failed to reverse cold hypersensitization, repeated intranasal administration significantly alleviated cold hypersensitization (Figure 2A). Additionally, the area under the curve analysis demonstrated that repeated administration of HU-308 produced a significantly greater reduction in cold hypersensitization in IONC mice compared with a single administration (Figure 2B,C).

### 3.2. Oral Administration of CB_2_ Agonist Does Not Affect Cold Hypersensitization in IONC Mice

To evaluate the therapeutic potential and use of intranasal administration for PTTN, we investigated whether a repeated oral administration with an equivalent dose of HU-308 can alleviate cold hypersensitization. Although no significant changes were observed in the time course of drug administration (Figure 3A), a comparison of the area under the curve values revealed that repeated oral administration of HU-308 resulted in a slight but significant improvement in cold hypersensitivity in IONC mice, similar to the effects observed with the intranasal administration (Figure 3B). The area under the curve analysis indicated that repeated intranasal administration of HU-308 produced a significantly greater amelioration of cold hypersensitization compared with repeated oral administration, suggesting a superior therapeutic effect of the intranasal route (Figure 4).

### 3.3. Intranasal Administration of CB_2_ Agonist Alleviates Sp5C Microglial Activation in IONC Mice

Given the established role of microglial activation, as evidenced by increased Iba-1 expression, in the pathogenesis of IONC-induced hypersensitivity [22], we examined the effects of HU-308 treatment on microglial activation in Sp5C. In the Sp5C region, IONC mice exhibited a significant elevation in the number of Iba1-positive cells, Iba1 fluorescence intensity, and cellular area compared to sham mice. The intranasal administration of HU-308 demonstrated a trend toward reducing these parameters (Figure 5).

### 3.4. Pretreatment with CB_2_ Antagonist Inhibits the CB_2_ Agonist-Induced Anti-Nociceptive Effect in IONC Mice

It is necessary to ascertain whether the analgesic effect of HU-308 is a CB_2_-mediated response. It has previously been reported that the analgesic efficacy of HU-308 for formalin-induced peripheral pain was attenuated by pretreatment with CB_2_ antagonist SR 144528 [25]. Similarly to previous findings, pretreatment with SR 144528 15 min prior to each HU-308 intranasal administration reduced the effects of HU-308 on cold hypersensitivity and the microglial activation in Sp5C (Figure 6 and Figure 7), suggesting that the analgesic effects of HU-308 are mediated by CB_2_ activation in IONC mice.

## 4. Discussion

The present study explored the therapeutic efficacy of CB_2_ receptor activation in a preclinical model of PTTN. Repeated intranasal administration of HU-308, a selective CB_2_ agonist, significantly alleviated acetone-induced cold hypersensitivity and attenuated microglial activation in IONC mice. These findings provide compelling evidence supporting the therapeutic potential of CB_2_ agonists for the treatment of PTTN.

Numerous studies have demonstrated that the aberrant activation of microglia in the spinal dorsal horn or the Sp5C region is pivotal in the pathogenesis of neuropathic pain [13,14,15,16,17]. Microglia contribute to chronic pain conditions by inducing changes in neuronal plasticity, including long-term potentiation, through the release of inflammatory cytokines [15]. Given these findings, targeting microglia with anti-inflammatory drugs presents a compelling strategy for the development of novel therapeutics for neuropathic pain. CB_2_ proteins, expressed on immune cells including microglia, have been shown to mediate anti-inflammatory effects [29]. Previous studies have indicated that CB_2_ activation can significantly suppress inflammatory responses induced by various stimuli in microglia [30,31,32,33,34]. For instance, CB_2_ activation has been shown to suppress the Janus kinase (JAK)/signal transducer and activator of transcription 1 (STAT1) pathway induced by interferon-gamma and the mitogen-activated protein (MAP) kinase pathway triggered by Toll-like receptor 4 activation, leading to the decreased production of inflammatory cytokines and exertion of anti-inflammatory effects [30,31]. In the present study, while a single administration of HU-308 failed to alleviate cold hypersensitivity in IONC mice, repeated administration produced a significant improvement. Additionally, the significant attenuation of the therapeutic effects by pretreatment with SR 144528, a CB_2_ antagonist, strongly supports the notion that HU-308 exerts its therapeutic effects via CB_2_. These findings suggest that the transient stimulation of CB_2_ may be inadequate to counteract the multifaceted effects of inflammation, and that sustained CB_2_ activation and its associated anti-inflammatory effects are essential for ameliorating pain conditions. The observation that only repeated administration of HU-308 demonstrated therapeutic effects can be attributed to the fact that inflammatory responses form a feed-forward loop, perpetuating the pathological state. A previous study has also demonstrated that minocycline, a microglial inhibitor, exhibited greater efficacy with repeated administration compared with a single dose in the treatment of oxaliplatin-induced mechanical allodynia [35]. Thus, the transient inhibition of microglial activity may be insufficient to achieve adequate therapeutic outcomes, because microglia were continuously activated before the treatment and sustained inflammation; hence, the continuous suppression of microglia may be essential. Another possibility is that repeated administration of HU-308 enabled its accumulation in brain parenchyma, reaching effective concentrations. Although the details of HU-308 accumulation in brain tissue are unknown, the calculated logarithm of the octanol–water partition coefficient (cLogP) of 8.97 indicates that it is highly lipophilic and may accumulate in the lipid-rich central nervous system [36].

Intranasal administration offers a non-invasive route for drug delivery to the central nervous system, specifically targeting the olfactory bulb and the pons and the medulla oblongata via the olfactory and trigeminal nerve pathways, respectively [19,20,21]. This is supported by studies demonstrating the accumulation of radiolabeled antibodies within these brain regions after intranasal administration [20]. The current study demonstrated in IONC mice that repeated intranasal administration of HU-308 attenuated microglial activation in Sp5C, a brain region located around the pons and the medulla oblongata. Given the analogous roles of Sp5C and the spinal dorsal horn in pain processing and the established role of microglial activation in neuropathic pain [14,15,16,17], we propose that HU-308 exerts its analgesic effects by targeting CB_2_ on Sp5C microglia. However, considering that intranasal administration can deliver drugs directly to the brain via the trigeminal nerve pathway, it is conceivable that HU-308 exerts its therapeutic effects by interacting with the damaged nerve or the trigeminal ganglion during its transit [19]. Previous studies have shown that macrophage infiltration and accumulation in the trigeminal ganglion following nerve injury play a crucial role in the pathogenesis of PTTN [37]. Additionally, CB_2_ activation has been reported to inhibit tumor necrosis factor-α-induced inflammatory responses and monocyte migration [38]. Considering these findings, it is plausible that the activation of CB_2_ on macrophages accumulated in the trigeminal ganglion alleviates chronic pain in PTTN by suppressing inflammation and reducing cellular infiltration. Further studies are required to elucidate the mechanisms underlying the analgesic effects of CB_2_ activation in PTTN.

In this study, intranasal administration was used to deliver drugs to the medulla oblongata, the injured trigeminal nerve, and the trigeminal ganglion, which are target tissues for the treatment of PTTN. The intranasal administration of HU-308 ameliorated cold hypersensitization and suppressed the abnormal activation of Sp5C microglia in IONC mice. In contrast, repeated oral administration of HU-308 resulted in only a slight improvement in cold hypersensitivity in IONC mice. While oral administration is a convenient method that enables widespread distribution of drugs throughout the body, it has several drawbacks, including the requirement for larger drug doses compared with local administration and the potential for unexpected systemic side effects. Additionally, CB_2_ is expressed not only on immune cells but also in some parts of the gastrointestinal tract [39]. Therefore, achieving sufficient analgesic effects with oral administration may require higher doses of HU-308, increasing the risk of inducing unexpected systemic side effects. These results suggest that the intranasal administration of CB_2_ agonists may be a promising therapeutic approach for the treatment of PTTN, not only in terms of therapeutic efficacy but also in terms of avoiding side effects and cost-effectiveness.

## 5. Conclusions

Neuropathic pain, characterized by its resistance to conventional analgesics, remains a significant clinical challenge. In this study, we used IONC mice, a well-established animal model of PTTN, to investigate the therapeutic potential of a selective CB_2_ agonist. Remarkably, repeated intranasal administration of HU-308 significantly attenuated pain-related behavior and pathological changes. These findings strongly support the hypothesis that CB_2_ may be a novel therapeutic target for the treatment of trigeminal neuropathic pain, and intranasal delivery may offer a promising approach for targeting the Sp5C microglia.

## Figures and Tables

**Figure 1 cells-13-01943-f001:**
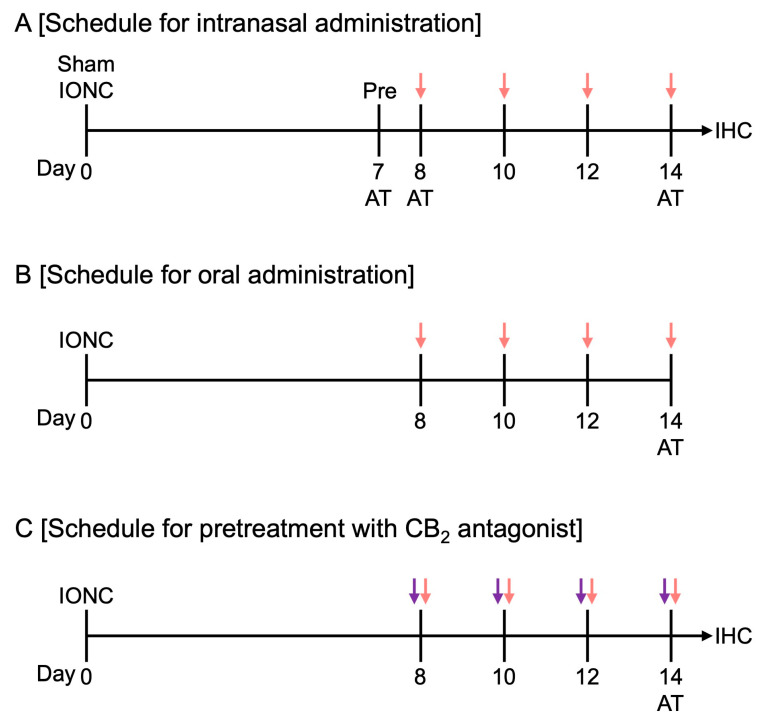
Drug administration and behavioral experiment schedule. (**A**) Intranasal administration of HU-308 (30 nmole, 10 µL) was performed 8, 10, 12, and 14 days after nerve injury. The red arrows indicate when HU-308 was administered. An acetone test (AT) was conducted one day before the initial drug administration (day 7), immediately after the initial administration (day 8), and after repeated administrations (day 14). Following the behavioral tests (day 15), brain tissue samples were collected from the mice for immunohistochemical analysis (IHC). (**B**) Oral drug administration was performed 8, 10, 12, and 14 days after nerve injury. The red arrows indicate when the drug was added. An acetone test (AT) was conducted after repeated administrations (day 14). (**C**) Pretreatment with SR 144528 was intranasally administered 15 min before each HU-308 administration. The red and purple arrows indicate when the HU-308 and SR 144528 were administered, respectively. An acetone test (AT) was conducted after repeated administrations (day 14).

**Figure 2 cells-13-01943-f002:**
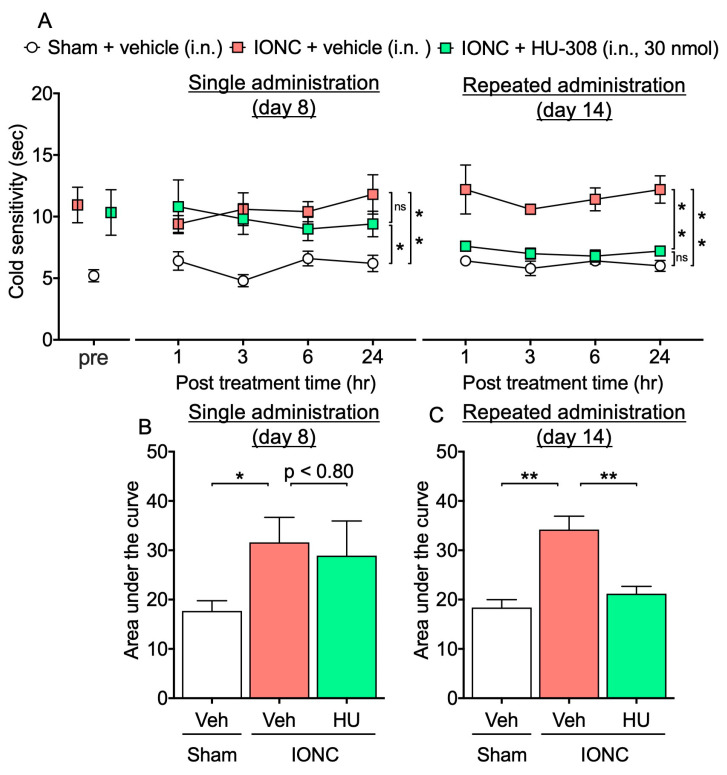
Effect of intranasal administration of HU-308 on cold hypersensitization in IONC mice. (**A**) The responsiveness to acetone was assessed at 1, 3, 6, and 24 h following a single and repeated intranasal administration of HU-308 (30 nmole, 10 µL, i.n.). The area under the curve was determined from the acetone test data following either single (**B**) or repeated (**C**) intranasal administration and compared. Individual data and mean ± SEM are shown. N = 5 (A); * *p* < 0.05, ** *p* < 0.01; n.s., not significant (two-way RM ANOVA followed by Tukey’s multiple comparisons test). (**B**,**C**) * *p* < 0.05, ** *p* < 0.01 (one-way ANOVA followed by Tukey’s multiple comparisons test).

**Figure 3 cells-13-01943-f003:**
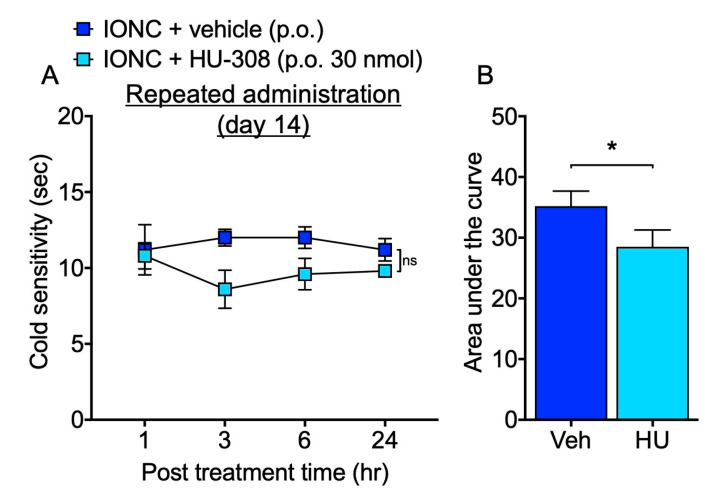
Effect of oral administration of HU-308 on cold hypersensitization in IONC mice. (**A**) The responsiveness to acetone was assessed at 1, 3, 6, and 24 h following repeated oral administration of HU-308 (HU, 30 nmole, 100 µL, p.o.). (**B**) The area under the curve was determined from the acetone test data following repeated oral administration and compared. Individual data and mean ± SEM are shown. N = 5 (**A**); n.s., not significant (two-way RM ANOVA followed by Sidak’s multiple comparisons test). (**B**); * *p* < 0.05 (Unpaired *t*-test).

**Figure 4 cells-13-01943-f004:**
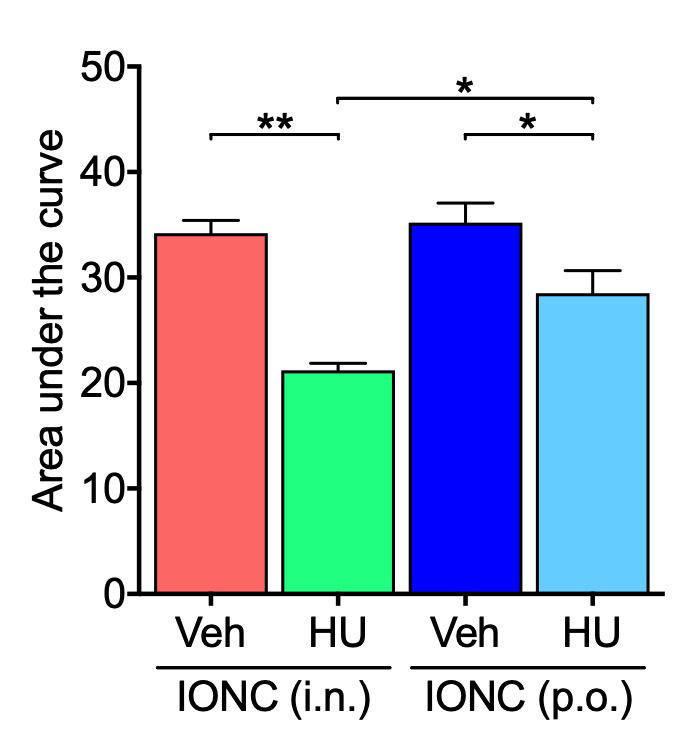
Comparative analysis of the effects of intranasal and oral administration of CB_2_ agonist on cold hypersensitivity in IONC mice. The area under the curve was determined from the acetone test data following either intranasal (i.n.) or oral (p.o.) repeated administration of HU-308 (HU, 30 nmole, 10 µL) and compared. Individual data and mean ± SEM are shown. N = 5; * *p* < 0.05, ** *p* < 0.01 (two-way ANOVA followed by Tukey’s multiple comparisons test).

**Figure 5 cells-13-01943-f005:**
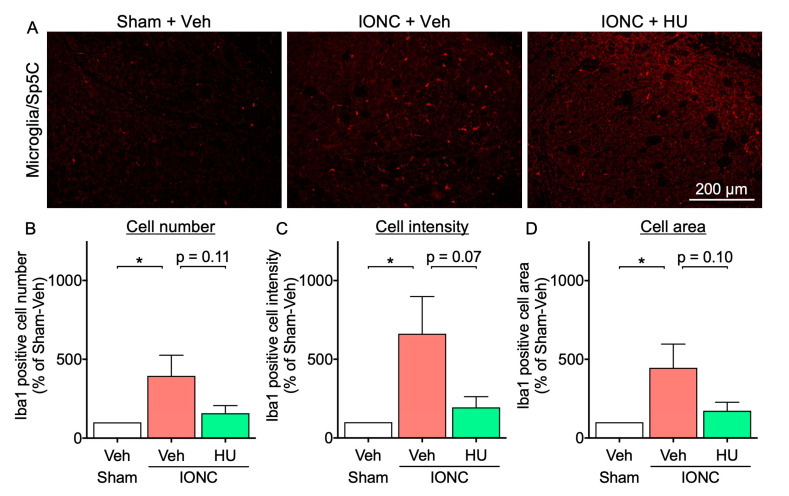
Effect of intranasal administration of HU-308 on Sp5C microglial activation in IONC mice. (**A**) Fluorescent photomicrographs of Iba-1 in Sp5C from sham and IONC mice following repeated intranasal administration of HU-308 (HU, 30 nmole, 10 µL). Scale bar = 200 µm. The number of Iba1-positive cells (**B**), the Iba1 intensity (**C**), and the cell area (**D**) were calculated from N = 5 mice. Individual data and mean ± SEM are shown. * *p* < 0.05 (one-way ANOVA followed by Dunnett’s multiple comparisons test).

**Figure 6 cells-13-01943-f006:**
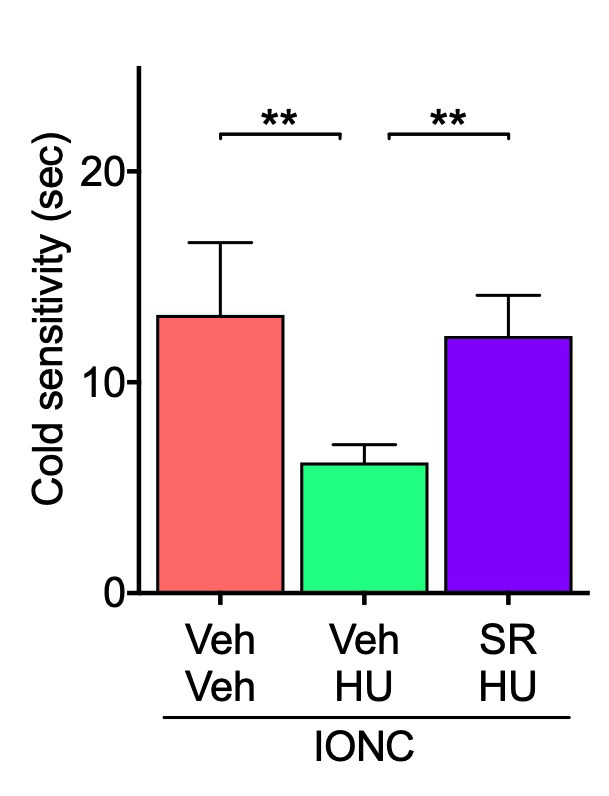
Effect of pretreatment with SR144528 on HU-308-inhibited cold sensitization in IONC mice. The responsiveness to acetone was assessed 3 h following repeated intranasal administration of HU-308 (HU, 30 nmole, 10 µL). SR 144528 (SR, 100 nmole, 10 µL) was administered 15 min before HU-308. Individual data and mean ± SEM are shown. N = 5, ** *p* < 0.01 (one-way ANOVA followed by Dunnett’s multiple comparisons test).

**Figure 7 cells-13-01943-f007:**
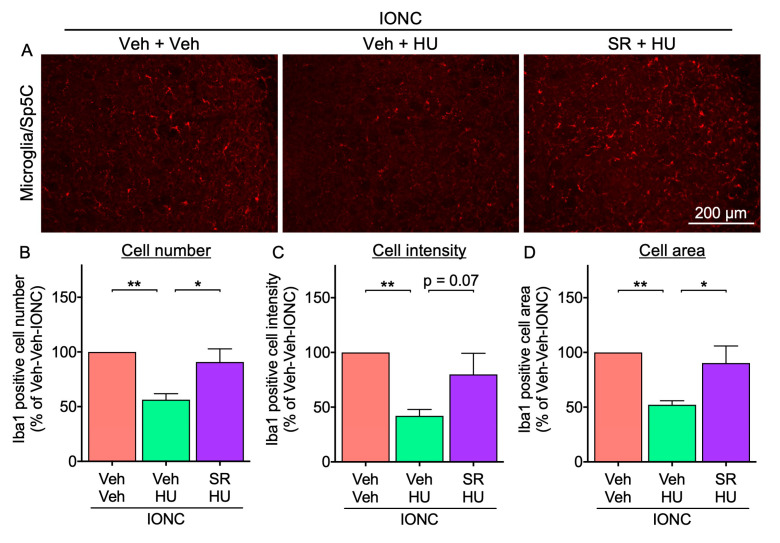
Effect of pretreatment with SR 144528 on HU-308-inhibited microglial activation in IONC mice. (**A**) Fluorescent photomicrographs of Iba-1 in Sp5C from IONC mice were assessed 3 h following repeated intranasal administration of HU-308 (HU, 30 nmole, 10 µL). SR 144528 (SR, 100 nmole, 10 µL) was administered 15 min before HU-308. Scale bar = 200 µm. The number of Iba1-positive cells (**B**), the Iba1 intensity (**C**), and the cell area (**D**) were calculated from N = 5 mice. Individual data and mean ± SEM are shown. * *p* < 0.05, ** *p* < 0.01 (one-way ANOVA followed by Dunnett’s multiple comparisons test).

## Data Availability

The data supporting the findings of this study are available from the corresponding author upon reasonable request.

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
