# Peer review of "Intranasal Treatment with Cannabinoid 2 Receptor Agonist HU-308 Ameliorates Cold Sensitivity in Mice with Traumatic Trigeminal Neuropathic Pain"

_cells, 2024, doi:10.3390/cells13231943_

Round 1

Reviewer 1 Report

Comments and Suggestions for Authors

Posttraumatic trigeminal neuropathy (PTTN) is a sensory abnormality caused by injury to the trigeminal nerve during orofacial surgery. However, there is currently a lack of effective treatment. This current study found that continuous stimulation of CB2 ameliorates PTTN induced pain via inhibition of microglial activation base on the behavior change of PTTN mice. Therefore, it provides a potential candidate for novel therapeutic agents against PTTN. However, lack of molecular biological validation is the major flaw of the study. Major revision must be done before this manuscript could be accepted for publication.

1. Oral administration has a first-pass effect. Only one dose of oral administration does not mean that oral administration has no effect. It is recommended to try oral administration with different drug concentrations.

2. It is recommended that the description of the administration method in the methodology be more detailed. How to confirm the effective dose after nasal administration? Because mice have sneezing or coughing reflexes during inhalation.

3. In the results section, it is recommended to add behavioral changes in cold hypersensitivity of mice in the modeling group and control group at different time points within 14 days. It is recommended to use molecular biology experiments to detect changes in CB1 expression after drug administration to confirm the effect of the drug. After the antagonist is administered, it is recommended to add immunohistochemistry results to detect microglial activation.

Comments on the Quality of English Language

It is recommended to polish the language before the manuscript is accepted for publication

Author Response

We are very grateful for the Reviewer’s comments concerning our manuscript. We have carefully considered each of the suggestions and have rewritten the manuscript accordingly. Changes in the revised manuscript are in red type.

Reviewer’s comment 1: Oral administration has a first-pass effect. Only one dose of oral administration does not mean that oral administration has no effect. It is recommended to try oral administration with different drug concentrations.

Response: Upon careful review of Reviewer 2's comments, we have identified a statistical error in our original analysis. Consequently, we have revised the manuscript to reflect a slight analgesic effect observed with oral administration of HU-308. We also hypothesize that increasing the oral dosage of HU-308 would enhance its analgesic effects. However, we interpret the results of the current study as indicating that intranasal administration is more effective than oral administration, if the same dose of HU-308 is administered by each route. Unlike oral administration, intranasal administration, which is expected to exhibit local effects at lower concentrations, is considered an effective administration method for post traumatic trigeminal neuropathy.

                Abstract: The therapeutic efficacy of oral HU-308 was found to be less pronounced in alleviating cold hypersensitivity in IONC mice compared to intranasal administration, indicating the potential advantages of the intranasal route. (Page 1, Line 21-23)

                Result section: Although no significant changes were observed in the time course of drug administration (Figure 3A), comparison of the area under the curve values revealed that repeated oral administration of HU-308 resulted in a slight but significant improvement in cold hypersensitivity in IONC mice, similar to the effects observed with the intranasal administration (Figures 3B). The area under the curve analysis indicated that repeated intranasal administration of HU-308 produced a significantly greater amelioration of cold hypersensitization compared with repeated oral administration, suggesting a superior therapeutic effect of the intranasal route (Figure 4). (Page 6, Line 195-202)

                Discussion section: In contrast, the repeated oral administration of HU-308 resulted in only a slight improvement in cold hypersensitivity in IONC mice. (Page 10, Line 377-379)

Reviewer’s comment 2: It is recommended that the description of the administration method in the methodology be more detailed. How to confirm the effective dose after nasal administration? Because mice have sneezing or coughing reflexes during inhalation.

Response: To prevent the backflow of the administered solution during intranasal administration, we reduced the fluid volume to 10 µL, which is lower than the typical dosage of 20-30 µL in this method (Correa et al, Proc Natl Acad Sci USA, 2023). Additionally, in this experiment, we confirmed that there was no leakage from the nose of mice after nasal administration. Furthermore, we have added the following sentence to the methods section;

                Method section: Considering the potential for drug reflux, the administration volume was decreased to 10 µL, which is lower than the typical volume of 20-30 µL [24]. Following intranasal administration, animals were carefully monitored to ensure that no drug reflux occurred prior to behavioral testing. (Page 4, Line 124-127)

Reviewer’s comment 3: In the results section, it is recommended to add behavioral changes in cold hypersensitivity of mice in the modeling group and control group at different time points within 14 days.

Response: The current study also conducted the acetone test on days 7 and 8, in addition to day 14. Compared to the Sham group, the IONC group showed a significant increase in response to acetone stimulation on all three days (Fig. 2A). The accompanying figure provides a summary of the acetone test outcomes for both sham and IONC groups.

Reviewer’s comment 4: It is recommended to use molecular biology experiments to detect changes in CB1 expression after drug administration to confirm the effect of the drug.

Response: Given the high selectivity of both CB2 ligands HU308 and SR 144528 used in the current (Hanus et al, Proc Natl Acad Sci USA, 1999; Rinaldi-Carmona et al, Curr Med Chem, 1999), we believe that the involvement of CB1 receptors in our findings is minimal. However, the expression of not only CB2 but also CB1 proteins in the trigeminal nerve following injury and potential changes in the expression of these receptors in the trigeminal nerve and surrounding cells after CB2 ligands administration are important considerations for future studies.

Reviewer’s comment 5: After the antagonist is administered, it is recommended to add immunohistochemistry results to detect microglial activation.

Response: We further investigated the effect of pre-treatment with SR 144528, a CB2 antagonist, on the inhibitory effect of HU-308 on Sp5C microglia activation. The results showed that the inhibitory effect of HU-308 on Sp5C Iba1 fluorescence intensity (a marker of microglia activation) tended to be attenuated by pre-treatment with SR 144528, and furthermore, the intensity of Iba1 was almost similar to that of vehicle group when SR144528 was pretreated (Fig. 7). These findings suggest that the inhibitory effect of HU-308 on Sp5C microglia is mediated by the CB2. Taken together, these results suggest that intranasal administration of HU-308 may alleviate cold allodynia by inhibiting Sp5C microglia activation through the CB2.

                Result section: Similar to previous findings, pretreatment with SR 144528 15 min prior to each HU-308 intranasal administration reduced the effects of HU-308 on cold hypersensitivity and the microglial activation in Sp5C (Figures 6 and 7), suggesting that the analgesic effects of HU-308 are mediated by CB2 activation in IONC mice. (Page 7, Line 277-280)

Reviewer 2 Report

Comments and Suggestions for Authors

Comments:

1) In figure 4 when comparing vehicle and drug there was a significant difference in the (p.o.) orally administered groups.  But in figure 3 when drug was administered orally there was no significant difference between vehicle and drug.  Please clarify.

2) In the mice that received antagonist was this completed in a separate group of mice?  Was the antagonist SR 144528 administered only once or repeatedly every time HU-308 was given during the 14 days?  Was this testing of the antagonist on day 14 post nerve cut?

3) How was the microglial signal quantitated.  Was this intensity per cell or intensity over a certain area minus the background.  Was the intensity measure over the multiple sections.  Please expand the methods for this measurement.

4) Please clarify why the trigeminal ganglia was not analyzed in these studies as it was also a likely location of the effect of drug.

5) Do intranasal drugs penetrate beyond the medulla to thalamic or sensory nuclei of the brain.

Author Response

We are very grateful for the Reviewer’s comments concerning our manuscript. We have carefully considered each of the suggestions and have rewritten the manuscript accordingly. Changes in the revised manuscript are in red type.

Reply to reviewer 2

Reviewer’s comment 1: In figure 4 when comparing vehicle and drug there was a significant difference in the (p.o.) orally administered groups.  But in figure 3 when drug was administered orally there was no significant difference between vehicle and drug.  Please clarify.

Response: Thank you for your careful review. We apologize for the error in the statistical data presented in Figure 3. We have revised the figure accordingly. In addition, we have revised the relevant text as follows;

                Abstract: The therapeutic efficacy of oral HU-308 was found to be less pronounced in alleviating cold hypersensitivity in IONC mice compared to intranasal administration, indicating the potential advantages of the intranasal route. (Page 1, Line 21-23)

                Result section: Although no significant changes were observed in the time course of drug administration (Figure 3A), comparison of the area under the curve values revealed that repeated oral administration of HU-308 resulted in a slight but significant improvement in cold hypersensitivity in IONC mice, similar to the effects observed with the intranasal administration (Figures 3B). The area under the curve analysis indicated that repeated intranasal administration of HU-308 produced a significantly greater amelioration of cold hypersensitization compared with repeated oral administration, suggesting a superior therapeutic effect of the intranasal route (Figure 4). (Page 6, Line 195-202)

                Discussion section: In contrast, the repeated oral administration of HU-308 resulted in only a slight improvement in cold hypersensitivity in IONC mice. (Page 10, Line 377-379)

Reviewer’s comment 2: In the mice that received antagonist was this completed in a separate group of mice?  Was the antagonist SR 144528 administered only once or repeatedly every time HU-308 was given during the 14 days?  Was this testing of the antagonist on day 14 post nerve cut?

Response: The experimental protocol regarding SR144528 pretreatment has been updated, and a schematic representation of the experimental timeline is now presented in Figure 1C.

Method section: Experiments were conducted using groups of 5 mice to examine the effects of intranasal HU-308 administration (Figure 1A), oral HU-308 administration (Figure 1B), and pretreatment with SR 144528 prior to HU-308 administration (Figure 1C). (Page 2, Line 84-87)

SR 144528, which has been reported to abolish the effects of HU-308 [25], was intranasally administered 15 min before each HU-308 administration. (Page 4, Line 131-133)

                Figure 1 legend: (C) Pretreatment with SR 144528 was intranasally administered 15 min before each HU-308 administration. The red and purple arrows indicate when the HU-308 and SR 144528 was treated, respectively. The acetone test (AT) was conducted after repeated administrations (day 14).

Reviewer’s comment 3: How was the microglial signal quantitated.  Was this intensity per cell or intensity over a certain area minus the background.  Was the intensity measure over the multiple sections.  Please expand the methods for this measurement.

Response: We have revised the explanation of the fluorescence image analysis methods as follows;

Method section: Individual Iba1-positive cells in the images were identified using CellProfilerTM [26,27]. Briefly, for each cell, the cell number, the integrated fluorescence intensity of Iba1, and the average cell area were calculated using the MeasureObjectIntensity and MeasureObjectSizeShape module of Cell ProfilerTM. (Page 4, Line 154-158)

Reviewer’s comment 4: Please clarify why the trigeminal ganglia was not analyzed in these studies as it was also a likely location of the effect of drug.

Response: We also agree that the trigeminal ganglion plays a crucial role in the pathogenesis of PTTN, and intranasal administered drugs appear to reach trigeminal ganglion in the current study. Previous studies have demonstrated that the accumulation of macrophages in the trigeminal ganglion following trigeminal nerve injury, along with the exacerbation of inflammatory responses, is critical for the development of this condition (Batbold et al, J Neuroinflammation, 2017; Shinoda et al, Front Mol Neurosci, 2021). Additionally, it is well-established that macrophages express CB2 receptors (Tortora et al, Biomedicines, 2022). This paper marks the first to discuss the efficacy of intranasal administration with CB2 agonist for PTTN, and future studies will aim to elucidate the underlying mechanisms of disease improvement by delving deeper into the roles of not only Sp5C microglia but also macrophages within the trigeminal ganglion.

Reviewer’s comment 5: Do intranasal drugs penetrate beyond the medulla to thalamic or sensory nuclei of the brain.

Response: As discussed in the Discussion section (page 10, line 374-388), intranasal administration has been reported to deliver drugs and therapeutic antibodies from the periphery to the central nervous system via retrograde transport along the olfactory and trigeminal nerves. Previous studies have shown that intranasally administered radiolabeled antibodies can penetrate deep into the brain over time (Kumar et al, J Control Release, 2018).

Reviewer 3 Report

Comments and Suggestions for Authors

very intersting research, please provide as aditional supplementary file table with complete data, p values, effect sizes and post hoc pairwise comparisons

Author Response

We are very grateful for the Reviewer’s comments concerning our manuscript. We have carefully considered each of the suggestions and have rewritten the manuscript accordingly. Changes in the revised manuscript are in red type.

Reply to reviewer 3

Reviewer’s comment 1: Very intersting research, please provide as aditional supplementary file table with complete data, p values, effect sizes and post hoc pairwise comparisons

Response: We have included additional supporting materials related to the experimental data and statistical analysis.

Reviewer 4 Report

Comments and Suggestions for Authors

The manuscript by Morioko and colleagues presents timely and authoritative data on the efficacy of intranasal treatment with HU308 in alleviating cold sensitivity in mice during trigeminal neuropathic pain. The manuscript is clearly written, and the data provide an important update for the research community. The figures are well-constructed and offer valuable visual context, enhancing the overall clarity of the study.

The manuscript presents an update on the mechanisms behind the effectiveness of intranasal treatment. I have a few minor suggestions to further improve the manuscript:

Minor Suggestions:

  1. Materials and Methods Section:
    • Animals: Please include more details about the mouse line used in the study. Why were these particular mice chosen for your experiments?
  2. Infraorbital Nerve Cut Model as Posttraumatic Trigeminal Neuropathy in Mice:
    • Please provide a reference for the surgical protocol followed in this procedure.
  3. Acetone Test:
    • Kindly add a reference for the acetone test protocol.
  4. Figure 1:
    • It would be helpful to include specific details regarding the dosing regimen, such as the dose timing and the number of doses administered per day.
  5. Drug Treatment:
    • Could you explain the rationale behind the chosen dosage design? How was the concentration decided, and why was the approach of dosing after 14 days not attempted?
  6. Immunohistochemistry:
    • While IBA-1 is a well-known marker for microglia, it can also stain infiltrating macrophages. I recommend considering additional markers like CD68 to confirm your findings.
    • Please provide information on the number of sections and the number of mice used for quantification.
  7. Results:
    • Please clarify what is meant by "Repeated dose." Does this refer to multiple doses per day, or continuous dosing over a 14-day period?
  8. Figure 5:
    • The IBA-1 staining does not appear very clear. Could you specify the area of quantification? If possible, consider improving the quality of the images or using alternative images that better represent the results.

Author Response

We are very grateful for the Reviewer’s comments concerning our manuscript. We have carefully considered each of the suggestions and have rewritten the manuscript accordingly. Changes in the revised manuscript are in red type.

Reply to reviewer 4

Reviewer’s comment 1: Please include more details about the mouse line used in the study. Why were these particular mice chosen for your experiments?

Response: ddY mice, characterized by their gentle nature, have been widely employed in the field of pain research (Akimoto et al, Cell Death Dis, 2013; Tsubota et al, Eur J Pharmacol, 2020;). Our laboratory has published numerous studies on intractable pain using this mouse strain (Ma et al, Biol Pharm Bull, 2024; Yoshimoto et al, Exp Neurol, 2023; Hisaoka-Nakashima et al, Int Immunopharmacol, 2022; Kochi et al, Behav Brain Res, 2022; Sato et al, Biomed Pharmacother, 2022). Therefore, in this study, we investigated the effects of CB2 agonists on PTTN using ddY mice. Additionally, we have described the manuscript to include the Research Resource Identifiers (RRID) for this mouse strain (MGI:5558113) to provide more detailed information.

Reviewer’s comment 2: Please provide a reference for the surgical protocol followed in this procedure.

Response: We have described following reference to the experimental methods section for the surgical protocol.

                Ueta, Y.; Miyata, M. Brainstem local microglia induce whisker map plasticity in the thalamus after peripheral nerve injury. Cell Rep 2021, 34, 108823, doi:10.1016/j.celrep.2021.108823.

Reviewer’s comment 3: Kindly add a reference for the acetone test protocol.

Response: We have added references to the experimental methods section.

Kochi, T.; Nakamura, Y.; Ma, S.; Hisaoka-Nakashima, K.; Wang, D.; Liu, K.; Wake, H.; Nishibori, M.; Irifune, M.; Morioka, N. Pretreatment with High Mobility Group Box-1 Monoclonal Antibody Prevents the Onset of Trigeminal Neuropathy in Mice with a Distal Infraorbital Nerve Chronic Constriction Injury. Molecules 2021, 26, doi:10.3390/molecules26072035.

Trevisan, G.; Benemei, S.; Materazzi, S.; De Logu, F.; De Siena, G.; Fusi, C.; Fortes Rossato, M.; Coppi, E.; Marone, I.M.; Ferreira, J.; et al. TRPA1 mediates trigeminal neuropathic pain in mice downstream of monocytes/macrophages and oxidative stress. Brain 2016, 139, 1361-1377, doi:10.1093/brain/aww038.

Reviewer’s comment 4: It would be helpful to include specific details regarding the dosing regimen, such as the dose timing and the number of doses administered per day.

Response: We have added detailed explanations of the drug administration procedures to the experimental methods section.

Method section: Experiments were conducted using groups of 5 mice to examine the effects of intranasal HU-308 administration (Figure 1A), oral HU-308 administration (Figure 1B), and pretreatment with SR 144528 prior to HU-308 administration (Figure 1C). (Page 2, Line 84-87)

SR 144528, which has been reported to abolish the effects of HU-308 [25], was intranasally administered 15 min before each HU-308 administration. (Page 4, Line 131-133)

                Figure 1 legend: (C) Pretreatment with SR 144528 was intranasally administered 15 min before each HU-308 administration. The red and purple arrows indicate when the HU-308 and SR 144528 was treated, respectively. The acetone test (AT) was conducted after repeated administrations (day 14).

Reviewer’s comment 5: Could you explain the rationale behind the chosen dosage design? How was the concentration decided, and why was the approach of dosing after 14 days not attempted?

Response: Previous study demonstrated that intranasal administration of antibody results in approximately 0.15% brain penetration (Correa et al., Proc Natl Acad Sci USA, 2023). The extent of brain penetration for small-molecule drugs, however, is less well-established. Based on the previous finding that the mouse pons has a volume of approximately 20 mm3 (Badea et al, NeuroImage, 2017), we calculated that a 30 nmol dose of the compound, assuming a brain penetration rate of 0.1%, would yield an approximate local concentration of 1.5 µM in the peripontine region. Given that the Ki values of HU-308 for CB2 and CB1 receptors are 22.7 nM and >10 µM, respectively (Hanus et al, Proc Natl Acad Sci USA, 1999), we anticipate that this local concentration would be selective for CB2 receptors. In addition, given that an analgesic effect was observed at this dose and that this effect was blocked by a CB2 antagonist, we posit that the primary mechanism of action of HU308 involves CB2 receptor activation.

Reviewer’s comment 6: While IBA-1 is a well-known marker for microglia, it can also stain infiltrating macrophages. I recommend considering additional markers like CD68 to confirm your findings. Please provide information on the number of sections and the number of mice used for quantification.

Response: Thank you for your valuable comments. We agree that the infiltration of macrophages into the SP5C is a very interesting phenomenon. However, as you pointed out, CD68 is also expressed in macrophages (Khanduri et al, Front Immunol, 2023; Thenappan et al, Am J Respir Crit Care Med, 2010). Therefore, in future studies, we plan to investigate macrophage infiltration into the Sp5C by co-staining with Iba1 and the microglia-specific marker TMEM119.

Reviewer’s comment 7: Please clarify what is meant by "Repeated dose." Does this refer to multiple doses per day, or continuous dosing over a 14-day period?

Response: We have provided detailed explanations of the administration methods in the experimental methods section.

Reviewer’s comment 8: The IBA-1 staining does not appear very clear. Could you specify the area of quantification? If possible, consider improving the quality of the images or using alternative images that better represent the results.

Response: To quantify Iba1-positive cells, we performed image reanalysis using the automated image analysis software, CellProfilerTM. The integrated density of Iba1 within each identified Iba1-positive cell and the corresponding cell area were measured. Statistical comparisons were conducted to determine differences between groups.

Method section: Individual Iba1-positive cells in the images were identified using CellProfilerTM [26,27]. Briefly, for each cell, the cell number, the integrated fluorescence intensity of Iba1, and the average cell area were calculated using the MeasureObjectIntensity and MeasureObjectSizeShape module of Cell ProfilerTM. (Page 4, Line 154-158)

Result section: In the Sp5C region, IONC mice exhibited a significant elevation in the number of Iba1-positive cells, Iba1 fluorescence intensity, and cellular area compared to sham mice. Intranasal administration of HU-308 demonstrated a trend toward reducing these parameters. (Figure 5). (Page 7, Line 220-223)

Result section: Similar to previous findings, pretreatment with SR 144528 15 min prior to each HU-308 intranasal administration reduced the effects of HU-308 on cold hypersensitivity and the microglial activation in Sp5C (Figures 6 and 7), suggesting that the analgesic effects of HU-308 are mediated by CB2 activation in IONC mice. (Page 7, Line 239-242)

Round 2

Reviewer 1 Report

Comments and Suggestions for Authors

Accept the author's explanation and accept the revised content of the article